# A Comprehensive Evaluation of Studies on the Adverse Effects of Medications in Australian Aged care Facilities: A Scoping Review

**DOI:** 10.3390/pharmacy8020056

**Published:** 2020-03-31

**Authors:** Haider Qasim, Maree Simpson, Yann Guisard, Barbora de Courten

**Affiliations:** 1School of Biomedical Science, Charles Sturt University, Orange 2800, NSW, Australia; masimpson@csu.edu.au; 2Faculty of Science, Charles Sturt University, Orange 2800, NSW, Australia; yguisard@csu.edu.au; 3Faculty of Medicine, Monash University, Clayton 3168, VIC, Australia; barbora.decourten@monash.edu

**Keywords:** scoping review, ADR assessment, elderly, aged care facilities, medications monitoring, nursing home, drug review

## Abstract

**Aim:** this scoping review was designed to identify studies that assess adverse drug reactions (ADRs) for older people in Australian aged care facilities. This review critically evaluates each published study to identify the risk of, or actual, adverse drug events in older people. Inclusion criteria: this review considered any clinical studies that examined the adverse effects of medications in older people who were living in aged care facilities. This review considered qualitative studies, analytical studies, randomized controlled trials (RCTs), descriptive cross-sectional studies, and analytic observational studies that explored the use of medications and their adverse effects on older people in clinical settings (including aged care facilities). **Methods:** an initial search of the PubMed (United State National Library of Medicine), OvidSP, EBSCOHost, ScienceDirect, Wiley Online, SAGE, and SCOPUS databases, with full text was performed, followed by an analysis of the article’s title and abstract. Additionally, MeSH (Medical Subject Headings) was used to describe the article. The initial round of the database search was based on inclusion criteria from studies that assessed tools or protocols aiming to identify the adverse effects of medications on the elderly population suffering chronic conditions or multiple co-morbidities. Two reviewers screened the retrieved papers for inclusion. The data presented in this review are in tabular forms and accompanied by a narrative summary which aligns with the review’s objectives. **Results:** seven studies were identified, and the extracted data from these studies were grouped according their characteristics and the auditing results of each study. **Conclusion:** it would be beneficial to design a comprehensive or broadly adverse drug reaction assessment tool derived from Australian data that has been used on the elderly in an Australian healthcare setting.

## 1. Introduction

In 2050, there will be over one million Australians living in aged care facilities [1]. Approximately half of this population is predicted to have cognitive impairment, while the remainder are likely to suffer from one or more chronic diseases such as depression, diabetes, cardiovascular diseases, neurodegenerative diseases and rheumatological conditions [1]. The adverse effects of medications can complicate the management of multiple chronic diseases, which often makes it challenging to follow clinical guidelines [2]. Numerous tools and protocols are available to assess the side effects of medication in aged care facilities. However, they are often specific to certain medical conditions and do not provide a comprehensive assessment of the medication’s side effects [3]. Adverse drug reactions (ADRs), defined as unwanted harmful reactions, result from an intervention related to the use of one or more medicinal products [3] ADRs usually require prevention, specific treatment, the alteration of a dosage regimen, or drug discontinuation [4]. One of the major causes of ADRs arises from inappropriate prescribing cascades, whereby a new medication is given to manage the adverse effect of that inappropriate drug, thus exposing the patient to continued risks of ADRs from culprit drugs and the newly prescribed drug [5]. In some cases, the adverse drug reaction (ADR) symptoms may be incorrectly interpreted as a primary diagnosis rather than as side effects of the medications [6]. This added complication in distinguishing between drug-induced symptoms and definitive medical conditions may result in additional medications being prescribed [6]. 

A scoping review was selected over a systematic review because the concepts of a scoping review are ideal to determine the depth and breadth of a body of literature on a given topic, such as the adverse effect of drugs in older people; it also gives a clear indication of the volume of the literature and studies that are available on this topic and provides more detail to the focus. A scoping review is a useful approach to examine each piece of evidence in detail and concerns more specific questions and gives more illustration about the inclusion and exclusion criteria. A scoping review is applicable in our topic to identify: (1) the types of available evidence about the effects of medications on older people; (2) to clarify key concepts and definitions in the published papers; (3) to examine how the research or study was done or conducted on our topic; (4) to identify the characteristics of each included study; (5) and to identify and analyse the gap needing to be covered in clinical practice. 

This scoping review searched the current academic literature for ADR assessment in Australian aged care facilities to identify studies that were summarized, and a critical evaluation was undertaken for each study. We conducted our scoping review between 2017 and 2020, and papers published up to February 2020 were included, with MeSH terms updated to reflect narrower subheadings that were added since March 2017. Seven databases where searched: PubMed, OvidSP, EBSCOHost, MEDLINE, ScienceDirect, Wiley Online, SAGE, and SCOPUS. The inclusion criteria were established and were informed by the PICO model: No restrictions were set concerning the elderly in Australian health care (P), interventions/tools for monitoring (I), assessment of the adverse effects of medication tools (C) and medication management in nursing homes or hospitals (O) which were used to frame the data extraction. In addition to PICO, the following study selection criteria were formulated: randomized controlled trials (RCTs) and only full-length articles were considered for inclusion in this review. Two reviewers (HS and YG) independently selected titles and abstracts and the corresponding full text articles were included in this scoping review. Any discrepancies in judgment (whether the article was included or excluded) were discussed in order to reach a consensus (MS and/or BD) about final inclusion. 


**Aims and review’s questions**


The aim of this scoping review was to establish which tools or protocols are being used in Australia to determine the adverse effects of medications in older people living in Australian aged care facilities. More specifically, the review questions were: What are the types of adverse effects identification tools currently used in Australian health care settings (aged care facilities and hospitals)?What evaluation outcome measures have been reported for the tools in primary care settings?Does the tool or protocol minimise the adverse effects of medications without compromising the benefits of medications?Do the tools improve patients’ clinical outcomes by identifying inappropriate medication prescribed or medication errors?Do the tools or protocols support multi-disciplinary interventions through optimising day-to-day patient care?


**Inclusion criteria**


The selected studies were based on the following: The study was intended for patients aged 65 years or older;The study included older patients who experienced adverse effects of medications;The study included older patients suffering from the adverse effects of polypharmacy and living in aged care facilities or admitted in hospitals;The study investigated tools that were/are currently being used in Australia.


**Exclusion criteria (Round one)**


Studies were excluded if one or more of the following was determined: No data on the adverse effects;Study included paediatrics and we were unable to separately extract paediatric data;Duplicated studies;Study included only a single medication;Study population only included adults that were younger than 65 years;Primary objective was not the adverse effects of medications;Studies not in English;Studies focused on experimental medicines;Studies in phase III of a clinical trial.


**Exclusion criteria (Round two)**


Model designs of the study were insufficiently described;Validation of tests were ambiguous;Designs and measures were not detailed;It was indeterminate as to whether the measure has been accepted in practice.

## 2. Methods

We designed and conducted our scoping review by following guidelines published by the Preferred Reporting Items for Systematic Reviews and Meta-analysis (PRIMSA) [7]. 


**Search strategy**


The titles, abstracts, methods, results, discussion and finding for all published papers were screened by two reviewers against the agreed upon inclusion criteria. Disagreements between reviewers were resolved by further discussion of the reason for exclusion and a consensus was achieved. The search strategy and subsequent selection criteria of the identified published papers are displayed in Figure A1. A full search strategy for all databases is detailed in Appendix A. 


**Types of participants**


This review considered studies for the identification of the adverse effects that medications have had on older people in the primary care settings of Australia (regardless of whether the study was designed in Australia or overseas). Only studies that had their abstract in English were selected. There was no limitation in considering the date of acceptance for publication. 


**Concept**


This review explored and identified the characteristics of each study and critically measured their effectiveness on a patient’s health and wellbeing. Data from each study include: the number of participants in each study, drugs identified as contributing to major ADRs, rates of primary outcomes, drugs most frequently associated with outcomes, the most frequent body system affected by ADRs, acceptable low rates of loss to follow-up, binding outcome and potential bias.


**Context**


In this scoping review, no limit was set for the study setting or time frame. All studies, including the selected studies, were conducted in clinical settings (hospitals and nursing facilities). Table A1 is the summary of the selected studies.


**Information sources**


The database searches, up to February 2020, were obtained through PubMed, PMC, OvidSP, EBSCOHost, MEDLINE, ScienceDirect, Wiley Online, SAGE, and SCOPUS. Moreover, the searching strategy by Medical Subject Headings (MeSH) terms in popular and commonly used keywords and phrases was also obtained through PubMed. ScienceDirect and OvidSP searched for literature and dissertations, and abstracts were reached through SCOPUS. 


**Study selection**


The studies were identified through electronic databases and manual searches. A full set of the selected studies were exported from the databases into the reference manager software, EndNote X8 (Clarivate Analytics, PA, USA). Duplications were removed. Before formal screening and finalising the selection processes, a calibration exercise for the identified studies was performed by two reviewers (HQ and YG) independently. The purpose of this review was to refine the screening questions and to ensure consistency across reviewers for screening and to select eligible studies according to the inclusion criteria. Every article passed through a two-step process by two reviewers working independently: Step 1: the two reviewers (HQ and YG) screened all the titles and abstracts and they selected those that were relevant. Each reviewer independently assessed the article against the inclusion and exclusion criteria. The reason for exclusion was stated in EndNote. Step 2: after the abstract was selected, the full version of the selected article was retrieved and imported into EndNote. The two reviewers (HQ and YG) undertook a full review. Some studies were excluded after the selection. The reason for the exclusion of the full text review was noted in EndNote by each reviewer. The refined and retrieved articles for the review were compared by the two reviewers until the final agreed set was achieved. The disagreements between the two reviewers (HQ and YG) were resolved by mutual consensus discussion by a third co-author (MS or BD). None of the review authors were blinded to the journal titles, study authors, or institute where the article came from. The study selection process was determined and presented in the Preferred Reporting Items for Systematic Reviews and Meta-Analysis (PRISMA) flow diagram format. 


**Extraction and data presentation**


The data extracted from the included studies was based on the guides of the scoping review questions. The extracted data had been tabulated according to the author’s name, location, number of patients, number of drugs, rate of primary outcomes, drugs most frequently associated with outcomes, validation, most frequent body system affected by the adverse effects of medications, whether the selection was biased or not, acceptable low rate of loss to follow-up, and blinding outcome. Furthermore, the extracted data was audited and critically appraised by comparing data regarding their use in clinical practice, which health profession they were used by, if the study had been evaluated or not, which conditions were not used and why, and to determine if there were any limitations in practice. A summary table illustrating the audited and critically appraised data can be found in Table A2.

## 3. Results

The database searches yielded a total of 337 citations after duplicates were removed. The titles and abstracts for these 337 articles were screened by the first author, and 239 article titles and abstracts were excluded in round one due to having the following issues: the study had no data on adverse effects (42 articles), the study included paediatrics (24 articles), the study was duplicated (45 articles), the study used single medications only (51 articles), the study included participants younger than 65 years old (28 articles), the primary objective was not adverse effects of medication (31 articles), the study was not in English (seven articles), the study focused on experimental medicines (nine articles), and the study was in phase III of a clinical trial (two articles). The remaining 98 articles were considered for further detailed assessment of the full paper in round two, and 91 were excluded due to having the following issues: the model designs were not well described (52 articles), the methodologies were ambiguous (19 articles), the design and measures were not fully detailed (two articles), and the measure was not accepted in practice (18 articles). The search yielded a total of seven citations for inclusion in this review. 


**Outcome measured**


The seven studies that reported on the rate of adverse effects from prescribed medications in older people are summarized in Table A3. They identified which medications were involved in causing major adverse effects and worsened patient’s health conditions. However, none of these measures were able to predict the risk or rate of adverse drug effects to prevent health deterioration in older people. 

The cross-sectional study conducted by Harrison et al. [8] recruited 541 individuals from 17 residential aged care facilities around Australia. Of these, 82.8% were cognitively impaired and 64.3% were suffering from dementia. The objective of this study was to examine whether the DBI and Potentially Inappropriate Medications (PIM) were associated with quality of life in older people. This study was conducted with two instruments: the EuroQol Five-Dimensional Questionnaire (a measure of quality of life) and the Dementia Quality of Life Questionnaire. The results indicated that drugs with anticholinergic and sedative ADRs were associated with a lower quality of life [8]. 

Turner at al. [9] conducted a cross-sectional study to review the fall risk resulting from psychotropics and medications that cause orthostatic hypotension. This study involved 383 Australian older people whose medications were analysed with the Fall-Risk-Increasing Drugs (FRIDs) tool [9]. In comparison to older patients who were not frail, the outcome of this study identified that the risk of falls was underestimated or not recognised with respect to the contribution to risk for those drugs [9].

Inappropriate medication use is a common contributor to health deterioration in the elderly. Basger at al. [10] cross-referenced the treatment of common medical conditions in elderly people with the 50 highest-volume Australian Pharmaceutical Benefits Scheme (PBS) medications prescribed to Australians in 2006 [10]. The study found 96 cases that were not managed as effectively as they could be; 48 causes were dispensing overuse (e.g., too frequent use of medicines based on the prescribed dose). Eighteen patients who had a history of falls were not taking psychotropic medications (e.g., falls reported due to other medications rather than psychotropics, such as blood pressure medications). Nineteen patients with diabetes and cardiovascular events were not taking the recommended antiplatelet medicines or anticoagulants. Four patients taking non-steroidal anti-inflammatory drugs (NSAIDs) did not have pain. Three patients were taking additional selective serotonin re-uptake inhibitor (SSRI) together with other serotonergic effects and there were four cases of severe drug–drug interactions [10]. 

Ashoorian et al. [11] designed the My Medicines and Me Questionnaire (M3Q) as a self-reporting questionnaire for mental health patients who expressed concerns regarding side effects with their psychotropic medications [11]. A total of 205 older people from six mental health facilities were included. The results indicated that 77% reported sedation (a major risk of falls) and 23% reported gaining weight (a major risk of cardiovascular illness) [11].

The Modified Drug Adherence Work-Up Tool (M-DRAW) was developed by Lee et al. [12]. This tool has been designed to identify the barriers to medication adherence due to the side effects of medications. This tool asks the following “Do your medications give you side effects that make you NOT want to take it?” If so, further assessment of why the medications have side effects and the changed doses or changing medications will start from that point. This tool uses Likert scales for the responses (four-point scales of frequency) from one representing ‘never’ to four representing ‘often’. A pre- and post-interview design was established with a total of 172 participants. Based on their response, they were categorised into three adherence subdivisions: intentional non-adherence (INA), partial non-adherence (PNA), and adherers. Participants within INA and PNA groups were assigned to the intervention groups, while the adherer participants were assigned to the control group [12]. M-DRAW could provide recommendations to clinicians by giving them a systematic approach to overcome each identified barrier to adherence, especially non-adherence due to ADR [12]. 

McLeod et al. [13] developed a list of inappropriate prescribing practices for older people. The criteria were based on the following: prescriptions may introduce the patient to clinically significant risks of adverse effects, equally effective or more effective alternatives with less risk are available, and any clinical intervention that is reasonable enough to change the existing prescription to decrease morbidity [13]. The final list contained 71 inappropriate prescriptions for older people. Each item includes a clinical situation and each situation contains recommendations for alternative therapy and/or further investigations [13]. 

Finally, Nishtala and colleagues [14] conducted a drug burden index (DBI) study in 62 aged care facilities in New South Wales (NSW). DBI measures the effect of cumulative exposure to both anticholinergic and sedative medications on cognitive and physical functions in older adults [14]. DBI scores in older people were calculated, and the impact of the medication review on the DBI score after the uptake of pharmacist recommendations by GPs were evaluated. A total of 150,475 cases were collected (6751 cases including ADRs from psychotropic medications). The study determined and reported the neuropsychiatric adverse effects in older people [14]. 

## 4. Discussion

The current scoping review included a total of seven studies that met the inclusion criteria, so they investigated and described the adverse effects of the prescribed medication on older people by using tools or protocols designed for this purpose. For the Harrison study outcome, further studies would be suggested to examine whether deprescribing of medications included in the drug burden index (DBI) or Beers criteria may improve quality of life outcomes for these individuals, as well as to improve other consequences associated with reduced exposure to these medications, such as reduced hospitalisation and mortality [8]. 

Further studies are needed to establish the efficacy of the FRID tools and to rationalize or simplify medication regimens for elderly patients who are prescribed medications associated with orthostatic hypotension and psychotropics [9]. Further research will be required to determine whether de-prescribing fall risk-inducing medications will effectively reduce the risk of falls in older people [9]. 

Basger’s criteria was designed due to the deficiencies of the older Beers criteria in order to better suit the Australian health care system [10]. It is similar to Beers criteria, but it is a list of indicators based on, and derived from, Australian data sources rather than US sources [10]. The medications expressed in the collected sources have greater potential relevance in the Australian healthcare setting [10]. Additionally, it is developed from an analysis of the most commonly dispensed PBS medications and the most common conditions for which the elderly receive medical care [10].

M3Q has open-ended questions that elicit vital information about the patient’s adherence and evaluates the quality of life [11]. It allows the patients to communicate their feelings in writing by asking the patients to prioritise the most bothersome side effects. This instrument can also be used in metal health patients, but, in that case, the precision of the answers needs to be approved by nurses or doctors. As a result, it enables an action that would improve the therapeutic relationship with clinicians and improve adherence to prescribed medications [11]. Furthermore, this tool enhances clinician and patient communication and the capacity to work in partnership towards a common purpose. M3Q could be a subject of future investigations about variables that affect the patient’s perceptions and overall satisfaction with Australian heath care in a broader patient group [11]. 

The M-DRAW tool is acceptable and reliable to identify barriers to medication adherence and the causes behind non-adherence [12]. However, the validity of this tool is still uncertain, and further study needs to be done with a larger sample size and follow-up with patients [12]. 

The McLeod tool includes substantial information about the severity of the adverse effect of medications and rankings of the clinical importance of the medication risks [13]. The suggestions of alternative therapeutic options were based on the concept of more effective and less risky therapy [13]. This tool will help establish specific evidence-based guidelines for geriatric pharmacotherapy. Therefore, it would be advisable to revise the McLeod list of medications regularly, such as by further validation or validation in the Australian setting [13]. 

The findings by Nishtala’s study reinforce the importance of careful clinical assessment and management of older people who are at risk of increased anticholinergic burdens due to the use multiple neuropsychotropic drugs [14,15]. 

Generally, the idea of designing ADR assessment tools is essential at all stages of the medication management pathway. The designed tool needs to be derived from validated Australian data and be applicable to the Australian health care system. The designed tool needs to adopt the concept of multidisciplinary corporation, a structured approach to identify potential risks related to the risk of adverse effects of medicines and help to develop a framework for improvement strategies, and it can be a reliable resource to assist in reducing medication errors, overuse, and potential risky adverse effects. Thus, it may help clinicians to make the most appropriate clinical decisions for their patients.

## 5. Conclusions

To the best of our knowledge, numerous studies were done in Australia and overseas to assess the side effects of medication in older people. However, they are often specific to certain medical conditions and do not provide a comprehensive assessment of the medication’s adverse effects. This is of concern, given the increasing prevalence of age-related chronic diseases and associated disability, as well as the increasing number of Australians living in aged care facilities, leading to an increase in age-related disabilities and chronicity. Adverse medication-related incidents, unplanned medication related admission to hospital and inappropriate prescribing patterns are commonly observed in Australian elderly people. Moreover, these studies do not provide guidelines for alternative therapeutic options, nor do they provide recommendations that avoid interactions and ADRs. Therefore, it would be beneficial if Australian clinician researchers designed a predictive tool that integrates the information reported in this review to minimize the risks of ADRs.

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
