# Peer review of "A Comprehensive Evaluation of Studies on the Adverse Effects of Medications in Australian Aged care Facilities: A Scoping Review"

_pharmacy, 2020, doi:10.3390/pharmacy8020056_

Round 1

Reviewer 1 Report

Your paper deals with an important issue, adverse drug reactions (ADRs) among elderly people in Australia.  However, the research question and purpose of the paper are not clear.  While presented as a scoping review the application of formal, repeatable methods for the review is not in evidence.  The inclusion of "assessment ADEs tool" as comparator seems inappropriate.  Surely if the intervention is the use of a tool to monitor ADEs the comparator should be normal care, without the use of a tool. Table 3 is unclear.  For example, the column headed "Rates of primary outcome" contains fundamentally different kinds of results for the different cited studies.  Inadequate information is given on presentation of results (e.g. for the Nishtala study, are the results proportions, rates, or risks?)

Author Response

Thank you for your review, 

The answers to your questions as the following: 

NOTE: (I submitted the paper after an intensive correction to the English editing, and I think my writing is more acceptable for publication

1- The research question and purpose of the paper are not clear

Answer: In the new version, I added clear research questions, and I stated a clear purpose of the paper

2- The inclusion of "assessment ADEs tool" as comparator seems inappropriate. 

Answer: in the new version, I added inclusion and exclusion criteria clearly 

3- Table 3 is unclear

Answer: In the new version, I edited and revised the information in table 3 with my colleagues.

Now, the scoping review has been edited and massively improved. 

Reviewer 2 Report

This review of tools to address adverse drug effects among older Australians uses appropriate methodology and makes reasonable recommendations. Authors should spell out abbreviations when first introduces in the abstract as well as the body of the manuscript, and place a legend for each table which should stand alone regarding symbols and definitions.

Author Response

Thank you for your review, 

The answers to your questions as the following: 

1- Authors should spell out abbreviations when first introduces in the abstract as well as the body of the manuscript,

Answer: In the new version, I did precisely what you instructed. I spelt out the abbreviations. 

2- Place a legend for each table which should stand alone regarding symbols and definitions. 

Answer: Yes, In the new version, I did exactly what you instructed, I legend and links each table to their texts. For example, you may click at [Table 2] and the word document heading to directly to the appendix of table 2 at the end of the paper. 

NOTE: (I submitted the paper after an intensive correction to the English editing, and I think my writing is more acceptable)  

Reviewer 3 Report

Thank you for the opportunity to review this scoping review paper to address an important area of pharmacy practice. Overall, the paper is worthy of publication with some key changes needed to reach the required standard of a publishable scoping review.  I feel that the paper is not clear enough in its aims at the beginning, also not clear enough in the reporting of the criteria and why a scoping was chosen over a systematic review. There is also some confusion about the inclusion and exclusion criteria in terms of country of origin of studies, when the focus is on an Australian context and when / how these criteria were applied in the PRISMA flow diagram. Please address the following points:

1. The description in figure 1 states that the excluded studies were ‘poorly aligned’ with PICO. PICO is a tool that enables the researcher to consider the broad areas of interest, which then leads to the determination of search terms and specific inclusion and exclusion criteria. The screening stage should, therefore, have specific inclusion/exclusion criteria resulting in specific reasons why each study was included or excluded. For 239 studies it would be difficult to display these reasons on the flow chart, but perhaps these could be included in an appendix? This would help to demonstrate the rigour of the scoping review process.

2. The authors state that the PRISMA guidelines were followed. There are some good practice points that are included in the guidelines that have not been described in the paper, which would improve the article.

3. Abstract: The English could be improved as the points being made are unclear. In particular, the last sentence. All needs to change to 'nine'. Present in past tense. Add dates for the period that the scoping review was conducted.

4. Keywords - add 'scoping review'

5. Introduction: PRISMA-Scr suggests that the author should explain in the introduction why the review questions/objectives lend themselves to a scoping review approach. Please add an explanation for this with clear aims and objectives.

6. Methods: The first round of screening was very broad with no mention of what those screening criteria were. There is insufficient information to be able to replicate the study. What keywords/MeSH/search terms were used for the first round of screening? What were the full ‘must include’ criteria? For example,  the impression is given that in the first round of screening the population was worldwide i.e. any studies written in English and the second narrowed it down to just Australian studies. Is that actually the case? Was one initial screening criteria that all studies should include a tool for ADR assessment, or at this stage was it any study that mentioned ADRs? Were you looking at a particular study type or any experimental method? While a scoping review has more leeway than a systematic review in terms of adding additional screening criteria during the search process, there still needs to be specific inclusion and exclusion criteria for each stage. The inclusion criteria described in lines 65-68 seem to be for the second round of more focused searches.

7. Methods: Please add PICO in full.

8. Methods: Line 45 states that all studies assessed ADRs. Reading the descriptions of the studies, I am not sure this is the case. I haven’t read any of the studies, so I am basing the following comments solely on the text.

9. Line 67 - what does 'a validated data mean?'

10. An explanation of the concept and context would be useful earlier in the article.

11. I have concerns about including the Horne study based on the BMQ. This is measuring participants’ beliefs/perceptions about long-term side effects /ADRs, not whether they actually occurred? This study isn’t directly comparable with any of the others.

12. McLeod et al is based on inappropriate prescribing – does this necessarily lead to ADRs? It might, but I would argue that an ADR doesn’t always occur as a result of inappropriate prescribing. Is the intervention described in the summary table associated with this study actually reported on in the article, or is this work in progress? If it is not specifically reported on in the included study then I would question the relevance of this inclusion also.

13. Turner et al appears to look at fall risk, rather than ADRs per se.

14. The abstract mentions included studies identified risk of or actual ADR events, however this distinction isn’t referred to in the main body of the text. I wonder whether the scoping review would actually be more accurately described as identifying tools that assess factors that may lead to ADRs. This makes more sense when paired with the discussion, given that there is discussion about developing a tool that identifies ADR risk, not a tool to measure ADR rate.

15. Presentation - Table 3 - needs reformatting. Use past tense throughout. Needs careful proof reading - particularly the Discussion section.

Author Response

Thank you for your review, 

Your intensive reviews and instructions are precious and robust to my paper and make a massive improvement to my article. I want to present you a BIG THANK YOU for your enormous support. 

NOTE: (I submitted the paper after an intensive correction to the English editing, and I think my writing is more acceptable for publication

The answers to your questions as the following: 

1- In the new version, I specified intensively and the inclusion and exclusion criteria in round 1 and round 2. Furthermore, I determined precisely the reasons for exclusion in round 1 and round 2 (in the figure and the method section). Moreover, I explained in detail why I selected the scoping review rather than a systematic review. 

2- I applied and explained in the new version the details of PRISMA 

3- The abstract is already edited by English editors (I submitted the paper to the English Editor in this website, and they improve my writing a lot) 

4- I added the keyword (scoping review) 

5- In the newly edited version, I explained clearly the aim and the objective of my paper in detail

6- I explained the criteria of inclusion and exclusion of the first round, also create an appendix of how I searched through the database (by using MeSH searching items) - All details about that in appendix 1. Also, I specified precisely the reasons for exclusion in round 1 and round 2 (in the figure and the method section). 

7- I add the details of the PICO 

8- Table 1 and 2 assessed each selected study in details 

9- It was confusing words, I delete it, and I explained more 

10- In the new version, I explained in detail about the context and concept of my scoping review 

11- You correct, after the revision between my colleagues and me, we decide to delete the details of the BMQ - Horne study because it is irrelevant. 

12- Regarding McLeod study, the finds are representing the medications have been classified to the high risk to the elderly patients who admitted to hospitals or living in aged care facilities. This study always in progress and expanded continuously. The selected drugs in the study have a high potential of side effects and interacted with many other medications (the interactions almost causing additional adverse effects, especially for older adults suffering multiple-comorbidities). Furthermore, this study gives suggestions about what the alternative options instead of the listed harm medication. Therefore, for these reasons, McLeod study is essential to add in my scoping review. 

13- This tool is called: Fall risk-increasing drugs. In other words, this tool assesses the adverse effects of drugs have a potential risk of falls in the elderly. This study stated that psychotropic medications and medications causing orthostatic hypotension are highly potential risk of falls in older people. Therefore, this paper is an essential in my scoping review. 

14- The name and the purpose and the aim of the paper are changed according to your notes and instructions 

15- Table 3 has changed, and the discussion part of the manuscript was edited. 

Many thanks, 

Round 2

Reviewer 3 Report

The paper has been through a major revision in line with my comments. Thanks for the positive feedback. Some further minor corrections are as follows:

  1. Keywords - systematic review has been added instead of scoping review
  2. Line 44 - require or warrant - not both
  3. Line 64 - just to confirm was the scoping review conducted between 2017 and 2020 or were these the dates covered in the SR? I take it that it is the former, but this needs to be very clear. Maybe refer to both timescales. i.e. papers published up to February 2020 were included? See line 144 also.
  4. Line 74 - remove word 'systematic' from systematic scoping review - this is not a term - it is either one or the other.
  5. Line 78 - Add Aims and review questions (as these are not written as objectives). Maybe number the questions - these are great.
  6. Line 81 - past tense - 'review questions were...'
  7. I have not come across a round 1 and 2 approach to exclusion criteria before - but this makes sense. Is there some reference to support this approach?
  8. Line 115 - ditto re use of the term systematic scoping review - if this term is used, please let me know as I am not aware of it!
  9. Line 114 - is it a systematic review or a coping review?
  10. Line 177 again refers to a systematic review

Author Response

Dear Reviewer, 

Thanks so much of your reply and comments. I fixed all your comments as the following: 

  1. Keywords - I removed the systematic review and added the word: (scoping review)
  2. Line 44 - I kept (require) and I deleted the word (warrant)
  3. Line 64 - I clarified the following: (We conducted our scoping review between 2017 and 2020, and the papers published up to February 2020 were included...…) 
  4. Line 74 - I removed word 'systematic' from systematic scoping review, and I re-write it : scoping review 
  5. Line 78 - I removed the word (objectives) and I add (Aims and review questions) and the questions are numbered now. 
  6. Line 81 - I removed (are) and I add (were) as a past tense - 'review questions were...'
  7. The reference includes the round 1 and 2 is: ((Tricco AC, Lillie E, Zarin W, et al. PRISMA Extension for Scoping Reviews (PRISMA-ScR): Checklist and Explanation. Ann Intern Med. 2018;169:467–473. doi: https://doi.org/10.7326/M18-0850))
  8. Line 115 - I removed the word (systematic) and I kept (scoping review) 
  9. Line 114 - It is a (scoping review)
  10. Line 177 - It is a (scoping review) 

Thank you so much, you have a significance contribution to enhance and improve my paper for publication. I learnt a lot from you. I wish you were my supervisor. 

Thanks once again